# First Report of Septicaemic Listeriosis in a Loggerhead Sea Turtle (*Caretta caretta*) Stranded along the Adriatic Coast: Strain Detection and Sequencing

**DOI:** 10.3390/ani12182364

**Published:** 2022-09-10

**Authors:** Ludovica Di Renzo, Maria Elisabetta De Angelis, Marina Torresi, Valeria Di Lollo, Giovanni Di Teodoro, Daniela Averaimo, Sabrina Vanessa Patrizia Defourny, Federica Di Giacinto, Chiara Profico, Vincenzo Olivieri, Francesco Pomilio, Cesare Cammà, Nicola Ferri, Gabriella Di Francesco

**Affiliations:** 1Istituto Zooprofilattico Sperimentale (IZS) dell’Abruzzo e Molise “G. Caporale”, 64100 Teramo, TE, Italy; 2Centro Studi Cetacei Onlus, Centro Recupero e Riabilitazione Tartarughe Marine “L.Cagnolaro”, 65125 Pescara, PE, Italy

**Keywords:** *Listeria monocytogenes*, *Caretta caretta*, zoonotic disease

## Abstract

**Simple Summary:**

Listeria is a ubiquitous bacterium responsible for outbreaks in humans and animals. Therefore, marine monitoring and detection of this bacterium are important for public health. This work presents the first case of *Listeria monocytogenes* in a stranded loggerhead sea turtle (*Caretta caretta*) with a severe septicaemic infection. Gross and microscopic lesions showed chronic inflammation of all tissues due to septicaemia. Furthermore, a microbiological investigation confirmed the presence of *L. monocytogenes*. Whole-genome sequencing (WGS) was performed on isolated colonies for deep characterization of the strain, and antibiotic resistance genes were identified from the sequence.

**Abstract:**

Although there are increasing reports on the prevalence of *Listeria monocytogenes* in wild species, this is the first case of listeriosis in sea turtle. An adult female *Caretta caretta* was rescued after being stranded alive along the coast of the Abruzzo region (Italy) in summer 2021. The turtle died in 6 days due to respiratory failure. The necropsy showed widespread organ lesions, such as yellow foci of necrosis in many organs, gastrointestinal erosions, pericarditis, and granulomatous pneumonia. Microbiological and histological analyses were performed on several organs. *Listeria monocytogenes* was isolated from multiple organs, indicating a case of septicaemic listeriosis, and the genome was sequenced and characterized. All the colonies analysed belonged to the same strain serogroup IVb, ST388, and CC388.

## 1. Introduction

*Listeria monocytogenes (Lm)* is a facultative anaerobic Gram-positive coccobacillus ubiquitously distributed in a variety of ecological niches [1,2]. It is widespread within the environment, mainly in soil and decaying vegetation. Farm animal infection is usually caused by the ingestion of spoiled silage contaminated with *Lm*; moreover, wastewater and sewage are reported to be the cause of river contamination, explaining the circulation between water, plants, animals, and humans [3]. *Lm* is an intracellular pathogen responsible for several outbreaks in humans and animals. Indeed, it can infect a wide range of animal species. Isolation from mammals (rats, hedgehog, moose, otter, racoon, deer, bear, fox, and monkey) including humans has been reported several times [4,5,6], but infection in other animals, such as bird and fish species, is possible as well. Nevertheless, to date, only a few case studies of reptilian infections by *Lm* have been reported [7,8,9,10]. *Lm* can survive for a long time (up to 3 weeks) in a marine environment [11], where it is widespread among invertebrate and vertebrate organisms [12]. However, it has been isolated only twice in marine mammals [13,14] and never in marine reptiles.

Sea turtles, the most abundant marine reptile species in the Mediterranean Sea, play an important role as indicators of the marine ecosystem’s health. For this reason, constant monitoring is essential to better understand the spread of diseases, such as listeriosis, in the water ecosystem.

Currently, whole-genome sequencing (WGS) analysis is the best approach to characterize strains to perform source tracking and epidemiological investigation. Even more, strain typing can reveal important information about virulence, resistance, and antimicrobial genes of the causative agent of a disease. Above all the serotypes, 4b strains are usually linked to human clinical cases of listeriosis, and among them, Clonal Complex (CC)1, CC2, CC4, and CC6 are the most reported. Furthermore, CC1 usually causes rhomb-encephalitis and abortion in ruminants [15]. Various CCs have been isolated from different sources, and several attempts have been made to demonstrate a link between strain isolation and source adaptation. Even so, deeper investigations into the biodiversity of *Lm* are needed to understand the distribution of different clonal groups [16].

This work aimed to report the first case of *Lm* in marine turtles *C. caretta* in the Italian sea, characterize the isolated *Lm* strain, and find virulence genes and other genes responsible for its pathogenicity.

## 2. Case Description

An adult female loggerhead turtle (*Caretta caretta*) was rescued by the Regional Stranding Network of Abruzzo after being stranded along the coast of Ortona (CH) in June 2021 according to DG 21/167 of 2014 [17]. The sea turtle was promptly transferred to the Rescue and Rehabilitation Center in Pescara (Centro di Recupero e Riabilitazione di Tartarughe Marine “Luigi Cagnolaro”-CRTM) by the Centro Studi Cetacei Onlus (CSC), and its health status was thoroughly investigated. The turtle showed lethargy and acute respiratory distress and died after 6 days. The carcass was then transferred to the Istituto Zooprofilattico “G. Caporale” dell’Abruzzo e del Molise (IZS Teramo) for the necroscopic examination.

The sea turtle had a curved carapace length (CCL) of 77.5 cm and a body weight of 60.17 kg. The turtle was in poor body condition: it had sunken eyes and thin shoulders; the neck, axillary, and inguinal regions and the central plastron area were depressed. A significant bilateral purulent ocular discharge was observed. During necropsy, the turtle was identified as an adult female. The most significant post-mortem gross lesions included pericardial effusion (Figure 1A), yellow lamellate material and yellowish-grey warty buttons on the mucous membrane of the intestine (Figure 1B), and enlarged ovarian follicles and serosal fibrinous plaques of the urinary bladder (Figure 1C). Moreover, yellow foci of necrosis in the kidneys (Figure 2A), liver, and spleen (Figure 2B), as well as in the cardiac tissue (Figure 2C) with parenchymal flaccidity were observed. The lungs were severely compromised with bilaterally diffuse granulomatous lesions in the parenchyma.

Eye swabs, the heart, lungs, liver, kidneys, thyroid, spleen, liver, thymus, ovaries, bladder, pancreas, large and small intestines, a section of the brain, and a faecal sample were collected for microbiological investigations. Samples were sent to the National Reference Laboratory for *Listeria monocytogenes* (NRL-*Lm*) and were analysed according to ISO11290-1:2017 [18] with a modification regarding the identification method. Moreover, all samples were submitted for microbiological examination performed in sheep blood agar (IZS Teramo) after three days of incubation at 37 °C ± 1 °C in aerobic conditions.

Gram-positive bacteria were detected from all submitted organs. Five suspected colonies from each positive sample were selected and identified as *Listeria monocytogenes* with MALDI-TOF MS (MALDI Biotyper^®^, Bruker Daltonics Gmbh & Co. KG, Bremen, Germany). Isolated colonies were frozen stocked at −80 °C for the following DNA extraction and sequencing. 

For histological analyses, 2 µm of the brain, heart, lungs, spleen, kidneys, bladder, intestine, and spinal cord, previously collected in 10% neutral buffered formalin, were placed on glass slides, stained with hematoxylin and eosin (H&E) and Gram, and observed under the stereomicroscope. The histological findings were consistent with a systemic granulomatous disease. In particular, lesions were detected in several organs and were characterized by multifocal granulomatous pneumonia, myocarditis, splenitis, meningoencephalitis, nephritis, intense hyperaemia, and mucosal erosion of the urinary bladder and intestine (Figure 3A–E). In the lung, numerous Gram-positive bacteria were observed within granulomatous lesions (Figure 3F).

DNA extraction was performed on 5 out of 55 colonies among the ones isolated from the liver, brain, eyes, faeces, and spleen, using QIAamp^®^ DNA Mini Kit (Qiagen, Hilden, Germany) following the manufacturer’s protocol with minor modifications according to a previous study [19]. DNA quantity and quality were evaluated with a Qubit fluorometer (Thermo Fisher Scientific, Waltham, MA, USA) and Eppendorf BioSpectrometer fluorescence (Eppendorf s.r.l., Milano, Italy). DNA integrity was assessed with an Agilent 4200 TapeStation system (Agilent, Santa Clara, CA, USA). Starting from 1 ng of input DNA, the Nextera^®^ DNA Library Preparation kit was used for library preparation according to the manufacturer’s protocols. WGS was performed on the NextSeq^®^ 500 platform (Illumina^®^, San Diego, CA, USA) with the NextSeq 500/550 mid-output reagent cartridge v2 (300 cycles, standard 150 bp paired-end reads). For the WGS data analysis, an in-house pipeline [20] was used. The trimming step of raw reads was performed using Trimmomatic [21] and a quality control check of the reads using FastQC v.0.11.5 (https://github.com/s-andrews/FastQC ). De novo assembly of paired-end reads was carried out using SPAdes v3.11 **(**https://github.com/ablab/spades**)** [22] with default parameters for the Illumina platform 2 × 150 chemistry. Finally, the quality check of the genome assemblies was performed with QUAST v.4.3 (https://github.com/ablab/quast). All the genomes that met the quality parameters recommended by Timme et al. [23] were used for the subsequent analysis steps. Multilocus sequence typing (*MLST*), based on the Pasteur scheme, was used to characterize *Lm* strains and detect the sequence type (ST) and Clonal Complex (CC) querying the Pasteur Institute platform (https://bigsdb.pasteur.fr/listeria/ (accessed on 3 June 2022)). All the colonies analysed belonged to the same strain serogroup IVb, ST388, and CC388. 

Core genome MLST (cgMLST) of *Lm* was calculated according to the Institute Pasteur’s scheme of 1748 target loci using the chewBBACA allele-calling algorithm [24]. Genomes with at least 1660 called loci (95% of the full scheme) were considered. Genome assemblies were screened for the presence/absence of virulence, disinfectants/metal resistance, and antimicrobial resistance genes using different functions available on the BIGSdb-*Lm* platform (http://bigsdb.pasteur.fr/listeria (accessed on 3 June 2022). All strains isolated in different organs shared the same pattern; hence, 64 out of 93 virulence genes were present. Among them, genes primarily involved in virulence were found: Listeria Pathogenicity Island (LIPI)-1 (actA, *prfA*, *hlyA*, *mpl*, *plcA*, and *plcB*), Internalins (*inlA*, *inlB*, *inlC*, and *inlJ*) and LIPI-4 (LM9005581-7009, LM9005581-7010, LM9005581-7011, LM9005581-7012, LM9005581-7013, LM9005581-70014), *IntA*, *inlF*, and *vip*. A heatmap of virulence genes’ presence is reported in Figure 4. Out of 117 genes investigated, the only Listeria Genomic Island (LGI) found was LGI-2_LMOSA2310. All genes belonging to the *SigB* factor were present (lmo0889, lmo0890, lmo0891, lmo0892, lmo0893, lmo0894 lmo0895, lmo0896), while the only Stress Survival Islet (SSI) genes identified were lmo1799, lmo1800, and SSI1_lmo0447. Finally, the antibiotic resistance genes detected from the sequenced genome were *fosX*, *lmo0919*, *mprF*, *norB*, and *sul.*

## 3. Discussion

This report describes a case of septicaemia with systemic granulomatosis lesions in *Caretta caretta* characterized by conjunctivitis, endocarditis, myocarditis, pneumonia, and hepatitis. These lesions are similar to what is reported in the literature concerning the atypical clinical form of listeriosis in adult sheep [25]. As the first case of listeriosis in *C. caretta*, comparisons are not available in the literature; hence, we cannot affirm if this clinical picture represents the typical or atypical form in loggerhead sea turtles.

Strain characterization showed that the isolated *Lm* belonged to serogroup IVb, CC388, and ST388. Classification of *Lm* strain into a specific serogroup can help in understanding some properties of the isolates. Strains belonging to the IVb serogroup usually harbour the full length of the *inl*A gene encoding a protein for the attachment of *Lm* to the host cells [26] and are therefore over-represented in human and animal listeriosis cases. Serotype 4b, together with serotype 1/2a, were the main serotypes also in other wildlife studies, including surveys of red deer, wild boar, and wild birds [27]. It is known that IVb clones of *Lm* show a predisposition for specific environmental niches, and the isolation of pathogenic *Lm* in a marine animal confirms that wild environments promote the selection of invasive *Lm* strains and their further spread [16]. Even more, it corroborates that water environments such as marine ecosystems are important for the survival and spread of *Lm*. ST388 was considered a rare sequence type, previously associated mainly with clinical cases and foods, but not environmental and animal sources [16]. However, it was reported in wild boar and deer tonsils, and in 2019, it was the triggering factor of a major listeriosis outbreak in Spain [28]. ST388 was also reported in a recent Italian study [29]. The previous association of ST388 mainly with clinical cases is consistent with our finding in a stranded turtle, supporting the idea that wild environments provide opportunities for the selection of invasive *L. monocytogenes* clones adapted to surviving in wild animals [28].

Genome characterization for virulence genes is important to identify potentially virulent strains, and as in this case, the presence of specific virulence genes and Listeria Pathogenicity Islands (LIPIs) may explain the invasive forms of listeriosis [30]. In particular, genome characterization allowed showing the presence of major genes involved in *L. monocytogenes* pathogenesis, stress, and antimicrobial resistance. Listeria Pathogenicity Island LIPI-1 (actA, prfA, hlyA, mpl, plcA, and plcB), essential for invasion, intracellular growth, and further spread to adjacent cells, was present as in all haemolytic Listeria species [31]. *PrfA* is the transcriptional activator of *L. monocytogenes* virulence genes *hlyA* and *act*A, while *hlyA* and *plcA* are important for lysis of the primary single-membrane vacuoles and escape; actA allows intracellular mobility and cell-to-cell spread [32]. Furthermore, Internalins (inlA, inlB, inlC and inlJ), involved in the expression of invasion-associated surface proteins, were detected. Indeed, *inlA* and *inlB* are important for the internalisation of *L. monocytogenes* and host penetration by phagocytosis [33]. Note that the presence of LIPI-4 genes, usually reported in hypervirulent strains, is usually associated with high tropism for neuronal and placental tissue [30]. Adhesive and invasive genes such as *ami* and *aut* were missing, in agreement with Shi and colleagues [34], who detected the absence of these genes in *L. monocytogenes* strains belonging to serogroup IVb. *InlF*, *lntA*, and *vip* genes were present as well. These genes are important for host immune response modulation. *inl*F interferes with pro-inflammatory cytokine production, allowing *L. monocytogenes*’s survival in macrophages and early-stage infection [35]. *lntA* modulates the host’s INF-l-mediated immune response, and *vip* interacts with the host endoplasmic reticulum for cell invasion and signalling events during infection [32]. *Listeriolysin S*, representing LIPI-3, is responsible for *L. monocytogenes*’s interaction with gut microbiota and is usually associated with serotype 4b, but it was missing in our strain [34]. The presence of SigB factors explains the strain’s ability to survive the acidic gastric environment. These genes are responsible for transcription control of the general stress response regulon (GSR), which mainly protects *L. monocytogenes* from environmental stress, and it is also involved in *L. monocytogenes*’s virulence, regulating *inl*A and *inl*B expression [36]. Furthermore, analysis of Stress Survival Islet (SSI) and Listeria Genetic Island (LGI) showed the presence of *lmo1799*, a gene encoding a putative LPXTG protein [37], *lmo1800*, encoding the phosphatase LipA, which targets host cell components essential for promoting infection independently of the pathway [38], and *lmo0447* for glutamate decarboxylase (GAD), which plays a role in acid stress survival [39]. Finally, LGI-2_LMOSA2310 can transfer various accessory genes into diverse chromosomal locations [16].

The analysis of antibiotic resistance genes confirmed the presence of intrinsic antibiotic resistance to fosfomycin, quinolones, sulfamethoxazole, oxacillin, and cephalosporins (*fosX*, *lin*, *mprF*, *norB*, and *sul)*, as previously reported by others [40,41].

## 4. Conclusions

In conclusion, we report the first case of listeriosis in a hospitalized loggerhead sea turtle. Microbiological investigation confirmed that *Lm* was the principal cause of disease and death. Due to the nature of the event, this study represents the first opportunity where clinical observations along with anatomical and histopathological examinations were possible. Moreover, characterization of isolated *Lm* colonies confirmed the presence of a virulent strain. This study confirms that the marine environment promotes the selection of invasive *Lm* strains and their spread. Thus, efficient monitoring and detection of *Listeria* spp. is crucial for controlling its spread.

## Figures and Tables

**Figure 1 animals-12-02364-f001:**
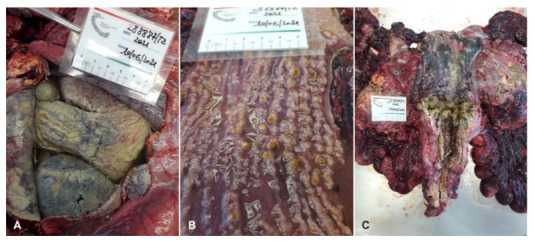
Pericardial effusion (**A**), yellow lamellated material and yellowish-grey warty buttons on the mucous membrane of the intestine (**B**), and enlarged ovarian follicles and bladder with serosal fibrinous plaques (**C**).

**Figure 2 animals-12-02364-f002:**
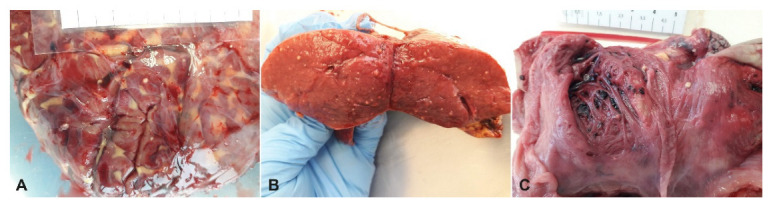
Yellow foci of necrosis in kidneys (**A**), spleen (**B**), and cardiac tissue (**C**).

**Figure 3 animals-12-02364-f003:**
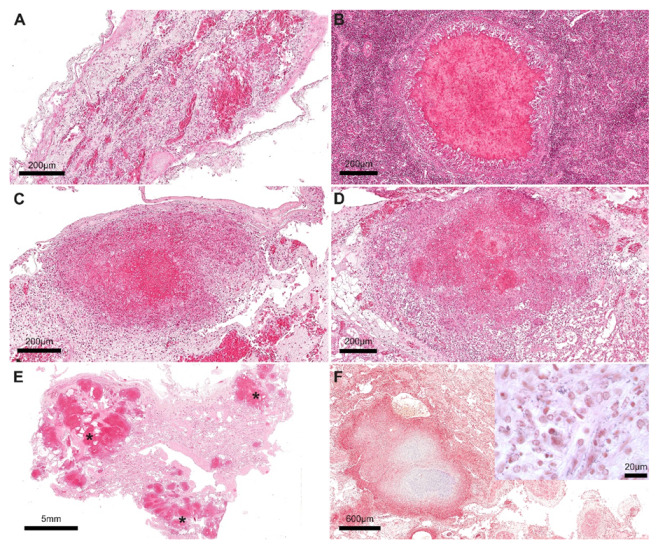
Granulomatous lesions with central necrosis were detected in the brain meninges (**A**), spleen (**B**), liver (**C**), and kidney (**D**). Multifocal granulomatous foci of bronchopneumonia were seen in the lung (black asterisks) (**E**). In the lung (**F**), numerous rod-shaped Gram-positive bacteria were detected between the inflammatory infiltrate (figure inset). Scale bar = 200 µm (**A**–**D**), 5 mm (**E**), 600 µm (**F**), 20 µm (F, inset). Hematoxylin and eosin stain: A–E. Gram stain: F and figure inset.

**Figure 4 animals-12-02364-f004:**
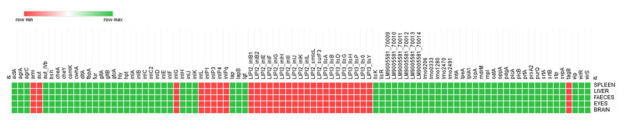
Heatmap virulence genes. In green, present genes; in red, missing genes.

## Data Availability

https://www.geocetus.it/ (accessed on 8 June 2021); genomic sequencing has been submitted to NCBI.

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
