# Peer review of "First Report of Septicaemic Listeriosis in a Loggerhead Sea Turtle (Caretta caretta) Stranded along the Adriatic Coast: Strain Detection and Sequencing"

_animals, 2022, doi:10.3390/ani12182364_

Round 1
Reviewer 1 Report
The authors reported a death case of loggerhead sea turtle (Caretta carretta) caused by Listeria monocytogenes (Lm) for the first time. The genomes of isolated Lm strains were sequenced and analyzed. These findings provide important clues for research on Lm biodiversity and pathogenicity. But there are some minor issues need improve.
1. In the abstract, the genomes data of isolated Lm were missing.
2. Line 91-94, Method description could be presented as support information, and provided microbiological investigations results here.
3. Line 95-98, Method description, not results.
4. Line 109-125, Method description, replace it with genomic data.
5. The genome accession number were missing.
6. Figure 3, including the scale bars were not clear, please zoom the pathological lesion areas.
7. Line 173, delete REF.
8. All species names like Listeria monocytogenes, should be written in italic.
9. Line 203, change "was" into "were".
Author Response
We would like to express our most sincere feelings of gratitude and appreciation toward the highly valuable and preciouse comments, remarks, and suggestions kindly made by this Reviewer. Wwe would like to kinldly inform you that we have carefully revised our "case report". Please see the attachment.

Reviewer 2 Report
Di Renzo et al. First report of septicemic listeriosis in a loggerhead sea turtle.
This case report describes the clinical, gross, and histological abnormalities in a fatal case of listeriosis in a loggerhead sea turtle, a potential first in this species. The authors additional provide detailed strain analysis as well as describe the virulence factors and antibiotic resistance genes present in the isolates. The manuscript is interesting and has great value as fatal listeriosis in this species would certainly be considered unusual and likely novel.
My major concern with this manuscript is the link between infection and disease. While I agree that with these lesions, obtaining these Lm isolates would strongly support Lm as the cause of mortality. However, I don’t think the bridge between infection and lesions was strong enough. Ideally, identifying and describing morphologically characteristic bacteria within areas of inflammation histologically would provide some support. A Gram stain at minimum or immunohistochemistry at best (which is commercially available) should be utilized to link bacteria to the lesion. Additionally, photomicrographs with higher magnification highlighting both the type of inflammation as well as bacteria (if visible) is warranted. If these additional modalities were pursued but failed to yield identification of the bacterium in tissue section, a discussion on possible reasons for the lack of identification should be included. The sentence starting in line 233 sums up my concerns with the manuscript. Microbiological investigation can absolutely confirm infection but does not confirm the cause of death. It supports listeriosis as the cause of death when combined with thorough gross and histopathological examination and compatible lesions.
Otherwise, my suggestions are largely to correct typographical errors or to consider rephrasing some sentences or sections to improve clarity. See below.
Minor suggestions
Line 19, 22, 25, and throughout: italics L. monocytogenes
Line 20, 26, and throughout: italics Caretta caretta
Line 25: rephrase; ‘Although there are increasing reports on…”
Line 28: ‘carcass necropsy’ and ‘organ lesions’ are redundant
Line 45: remove ‘,’ after ‘bird’
Line 50: consider rephrasing since sea turtles are widespread beyond the Mediterranean Sea
Line 51: run on sentence; consider making two sentences
Line 78: a reference stating that these morphometric values mean an adult female in poor body condition is encouraged
Line 79: remove ‘skinny’ as a lay term but also poor body condition was included in the line above
Line 79: ‘ocular’ should be the adjective before ‘discharge.’
Line 83: include ‘urinary’ if you are referring to ‘urinary bladder’ specifically
Line 89: remove ‘and’ before intestine. Also, which intestines (small or large)?
Lines 82 and 150: ‘button’ isn’t a term I hear much when referring to gross lesions. I suggest either ‘nodule’ or ‘button ulcer’ if the mucosa is, in fact, ulcerated.
Line 154: I don’t know if ‘parenchymal flaccidity’ can be assessed in these images and suggest removing the term here
Figure 3: While it is easy to tell that there is hemorrhage at this magnification, it is challenging to see the ‘granulomas’ or ‘granulomatous’ inflammation. I encourage removal of some of these images and include some with higher magnification to emphasize the type of inflammation present. The stain (assuming hematoxylin and eosin) should also be included.
Line 170: ‘granulomatosis disease’ may be confusing as it has a different connotation than the result of septicemia; consider rephrasing
Line 211: missing ‘.’ at end of the sentence
Lines 200-225: The detail on the virulence factors is interesting, but lengthy and not always directly applicable to this case. Consider consolidating and emphasizing how it is important to this case, sea turtle health or conservation, or listeriosis epidemiology rather than an encyclopedic list.
Author Response
We would like to express our most sincere feelings of gratitude and appreciation toward the highly valuable and precious comments, remarks, and suggestions kindly made by this Reviewer. We would like to kinldly inform you that we have carefully revised our "case report". Please see the attachment.

Round 2
Reviewer 2 Report
Di Renzo et al. septicemic listeriosis in a loggerhead sea turtle.
The updated manuscript is much easier to read and has an improved the diagnostic work-up. I recommend a few minor points below.
Line 50: remove ‘s’ from reptiles
Line 79: remove extra ‘.’
Lines 106, 161: A brief morphology modifier of ‘rod-shaped’ or ‘bacilli’ should be included to ensure the Gram profile and morphology aligns with L. monocytogenes.
Author Response
Dear Reviewer 2,
we would like to thank you for your precious advice. We are satisfied to know that in your opinion the paper has improved.
We have adjusted the paper following your last comments.
Please see the attachment.
Best regards
